# A cross-sectional pilot study of birth mode and vaginal microbiota in reproductive-age women

Christina A. Stennett[1,2], Typhanye V. Dyer[3], Xin He[3], Courtney K. Robinson[2], Jacques Ravel[2,4], Khalil G. Ghanem[5], Rebecca M. Brotman[1,2]*

1 Department of Epidemiology and Public Health, University of Maryland School of Medicine, Baltimore, Maryland, United States of America, 2 Institute for Genome Sciences, University of Maryland School of Medicine, Baltimore, Maryland, United States of America, 3 Department of Epidemiology and Biostatistics, University of Maryland, Maryland, United States of America, 4 Department of Microbiology and Immunology, University of Maryland School of Medicine, Baltimore, Maryland, United States of America, 5 Division of Infectious Diseases, Johns Hopkins University School of Medicine, Baltimore, Maryland, United States of America

* rbrotman@som.umaryland.edu

**Data Availability Statement:** The VM400 data can be found at the National Center for Biotechnology Information (NCBI) Database of Genotypes and Phenotypes (dbGaP) under accession number

## Abstract

Recent studies suggest that birth mode (Cesarean section [C-section] or vaginal delivery) is an important event in the initial colonization of the human microbiome and may be associated with long-term health outcomes. We sought to determine the association between a woman's birth mode and her vaginal microbiota in adulthood. We re-contacted 144 adult women from two U.S. studies and administered a brief survey. Vaginal microbiota was characterized on a single sample by amplicon sequencing of the V3-V4 hypervariable regions of the 16S rRNA gene and clustered into community state types (CSTs). We evaluated the association between birth mode and a CST with low relative abundance of *Lactobacillus* spp. ("molecular bacterial vaginosis" [Molecular-BV]) compared to *Lactobacillus*-dominated CSTs in logistic regression modeling which adjusted for body mass index, a confounder in this analysis. Twenty-seven women (19%) reported C-section. Overall, C-section showed a non-significant trend towards increased odds of Molecular-BV (aOR = 1.22, 95% CI: 0.45, 3.32), and *Prevotella bivia* was the strongest single taxa associated with C-section. However, because the two archived studies had different inclusion criteria (interaction p = 0.048), we stratified the analysis by study site. In the study with a larger sample size (n = 88), women born by C-section had 3-fold higher odds of Molecular-BV compared to vaginally-delivered women (aOR = 3.55, p = 0.06, 95% CI: 0.97–13.02). No association was found in the smaller study (n = 56, aOR = 0.19, p = 0.14, 95% CI: 0.02–1.71). This pilot cross-sectional study suggests a possible association between C-section and Molecular-BV in adulthood. However, the analysis is limited by small sample size and lack of comparability in participant age and other characteristics between the study sites. Future longitudinal studies could recruit larger samples of women, address the temporal dynamics of vaginal microbiota, and explore other confounders, including maternal factors, breastfeeding history, and socioeconomic status, which may affect the relationship between birth mode and vaginal microbiota.

phs001909.v1.p1. For the HCL study, all sequencing data are accessible using the NCBI Sequence Read Archive (SRA) BioProject accession PRJNA610195. The HCL survey data used in this study can be accessed directly from the Principal Investigator (Ghanem) and data release queries can also be referred to the Johns Hopkins Medicine Institutional Review Board (jhmeirb@jhmi.edu).

**Funding:** This study was funded by the National Institute of Allergy and Infectious Diseases (NIAID) U01-AI070921 (JR), NIAID R01-AI089878 (KGG), NIAID R01-AI119012 (RMB) and the National Institute on Aging T32-AG000262 (supporting CAS).The funders had no role in study design, data collection and analysis, decision to publish, or preparation of the manuscript.

**Competing interests:** The authors have declared that no competing interests exist.

## Introduction

There is emerging evidence that babies born by Cesarean section (C-section) are more likely to develop metabolic and chronic disorders, including celiac disease, diabetes mellitus, obesity, food allergy, and asthma, in early childhood, compared to those born by vaginal delivery.[1–7] In addition, the association between C-section delivery and obesity has been shown to persist into adolescence and adulthood.[8] Several of these chronic conditions have been associated with deviations in the colonization of the gut microbiota, with observed decreases in overall bacterial diversity and lack (or delayed colonization) of protective bacterial taxa among individuals born by C-section.[9–11] While evidence to suggest the influence of birth mode on the gut microbiota is building, there is little existing research on the impact of birth mode on the microbiota of the female reproductive tract.

Reproductive-age women with *Lactobacillus*-dominated vaginal microbiota are at lower risk for bacterial vaginosis (BV), which reduces the likelihood of sexually transmitted infection (STI) acquisition and development of abnormal pregnancy outcomes.[12] While there are a number of studies on demographic and behavioral risk factors for BV in adult women [13–16], less is known about the early risk factors for BV. We hypothesized that a woman's birth mode, that is, the method through which she was delivered by her mother, is an important early life factor in determining how a woman's vaginal microbiota is initially seeded and transitions into adulthood. Thus, any differences in the composition of vaginal microbiota attributable to birth mode must persist through known hormonally-driven transitions in the microbiota during early childhood and puberty, including the longitudinal dynamics observed among reproductive-age women in menstruation and pregnancy.

While some prior studies have found short-term differences in microbiota composition and structure at specific body sites between C-section- and vaginally-delivered neonates and infants, [17–19] others found that differences in infant microbiota weeks after delivery could not be explained by birth mode.[20,21] However, these studies did not assess the infant girls' vaginal microenvironments, and instead focused on the skin, oral, nasal, and/or gut microbiota. In addition, the infants were only observed for up to three years. Therefore, the purpose of this pilot study was to assess whether birth mode was associated with the composition of the vaginal microbiota of adult women in two study populations.

## Methods

We used cross-sectional, secondary data and repository samples from the Hormonal Contraception Longitudinal (HCL) Study [22] and the Vaginal Microbiota 400 Woman Study (VM400) [23]. A study coordinator re-contacted participants to query on additional questions related to birth mode and other confounding variables described below. In this study, women were asked to recall whether they were born vaginally or by C-section (i.e., how their mother gave birth to them) and were informed that the question was unrelated to how the adult participants gave birth to their own children.

### Secondary data sources

The HCL study aimed to evaluate how the initiation or cessation of hormonal contraceptives (HCs), including oral contraceptive pills, vaginal rings, and other methods, affect the vaginal microbiota and immune responses in the lower genital tract. Eligibility criteria included age (16 to 35 years old), not currently being pregnant, having a uterus (not post-hysterectomy), and not having any implanted uterine devices (such as Mirena or Paraguard). In addition, women diagnosed with illnesses that alter their immune system or hormone levels or who take medications that change their immune system or hormone levels were not eligible. This

analysis focused on a mid-vaginal swab that was collected by a clinician at the baseline visit. Between 2011 and 2016, the HCL study enrolled 125 women in the Baltimore, Maryland area.

Women in the VM400 study were recruited for a cross-sectional study at two sites in Baltimore, Maryland and one site in Atlanta, Georgia. Participants ranged from 12 to 45 years old, menstruated regularly, had a history of sexual activity (but none reported in the past 48 hours), reported no vaginal discharge, had not taken antibiotics or antimycotics in the past 30 days, and were not pregnant at the time of enrollment. Participants self-collected two mid-vaginal swabs using the ESwab system (Copan). The 396 participants recruited between 2008 and 2009 represented four racial/ethnic groups (white, black, Asian, and Hispanic) in roughly equal proportions.

## Processing of vaginal swabs

For the VM400 study samples, protocols for genomic DNA extraction from vaginal ESwabs (Copan) were carried out in 2009 as previously described.[23] The V3-V4 regions of the 16S rRNA gene were amplified by a one-step polymerase chain reaction (PCR) method.[24] Amplicon pooling, sequencing by Illumina MiSeq, and sequence data processing were conducted as described by Holm et al.[25]

For HCL vaginal samples, protocols were modified in year 2012, and genomic DNA was extracted from ESwabs with either the QS DSP Virus/Pathogen Midi Kit (Qiagen) on the QiaSymphony platform or with the MagAttract Microbial DNA Kit (Qiagen) using a custom automated protocol on the Hamilton Microlab Star if the samples performed poorly (<15,000 reads) in the first round of sequencing. For the QS DSP kit, samples were thawed on ice and a 500μl aliquot from the vaginal swab was used as input.[25] For the MagAttract kit, samples were thawed on ice and a 200μl aliquot from the vaginal swab was used as input for the kit following the manufacturer protocol. Cells were lysed by bead beating on the TissueLyser (Qiagen) at 20Hz for 20 minutes and the final elution volume was 110μl. The V3-V4 regions of the 16S rRNA gene were amplified by two-step PCR, with amplicon pooling, sequencing on an Illumina HiSeq 2500 instrument, and sequence data processing as previously described.[25]

For this analysis, amplicon sequence variants (ASVs) generated by DADA2 were taxonomically classified using the RDP Naïve Bayesian Classifier [26] trained with the SILVA v128 16S rRNA gene database [27] as implemented in the *dada2* R package.[28] ASVs of major vaginal taxa were assigned species-level taxonomy using speciateIT.[29] Taxa for which the total read count across all samples was less than $10^{-5}$ were regarded as likely contaminants and removed. Individual samples with fewer than 1,000 reads were also removed from the analysis. Vaginal microbiota were broadly clustered into five groups or "community state types" (CSTs) based on their diversity and relative abundance of bacteria.[23] For the purposes of this analysis, a binary outcome variable was created to contrast a low-*Lactobacillus* CST versus *Lactobacillus*-dominated states. CSTs dominated by *Lactobacillus*—*L. crispatus* (CST I), *L. gasseri* (CST II), *L. iners* (CST III), or *L. jensenii* (CST V)—were grouped together. CST IV, termed "Molecular-BV" [30], is a low-*Lactobacillus* state characterized by the abundance of BV-associated anaerobic organisms, including *Gardnerella*, *Megasphera*, *Sneathia*, and *Prevotella*. CSTs were assigned for each sample in this study based on similarity to the centroid of each CST as determined from a pool of over 13,000 archived vaginal samples.[31]

## Primary data collection

For the current study, the research team attempted to recruit all former HCL and VM400 study participants who had consented to re-contact for future research (over 450 women). Participants were contacted by phone or email and were given the option to complete a survey by phone or in

an online survey collected and managed with REDCap.[32] Informed consent was received for the current research prior to administering the survey. While a few participants were younger than 18 years when enrolled in the parent studies, all were older than 18 at re-contact. Participants had the opportunity to ask questions and decline further involvement at any time.

In the short survey, participants reported their birth mode (C-section or vaginal delivery), age at first menstrual period, and weight status at menarche. As the VM400 study did not collect height and weight information for body mass index (BMI) calculation, former VM400 participants provided their current height, current weight, and estimated weight at the time of parent study sampling. In addition, former VM400 participants were queried on breastfeeding history questions, which had been included in baseline surveys administered to HCL participants. There were many missing values for breastfeeding history questions with only 37.5% of VM400 participants providing responses. Behavioral and demographic data collected in the parent studies were also included in the analysis.

The institutional review boards at the University of Maryland School of Medicine and Johns Hopkins University provided approval for the protocol, including the re-contact of parent study participants.

## Statistical analysis

To compare distributions of selected health, behavioral, and demographic characteristics between women born by C-section and vaginal delivery, medians, frequencies, and percentages were calculated, and p-values were obtained using chi-square, Fisher's exact, or Wilcoxon rank sum tests. Logistic regression was used to examine the association between birth mode and Molecular-BV. As the two archived studies differed in participant characteristics and inclusion criteria, we planned *a priori* to assess potential effect modification due to study site. Several confounders identified in the literature as being associated with birth mode and/or Molecular-BV were considered for inclusion in the final adjusted model, including race/ethnicity, age, body mass index (BMI) at the time of sampling, and age and weight status at the first menstrual period. Because *L. iners*-dominated CST III is often associated with transitions to a Molecular-BV state [33], we conducted sensitivity analyses in which CSTs III and IV were each individually compared to the other *Lactobacillus*-dominated states (CSTs I, II, and V) in a multinomial model. These analyses were conducted using STATA/SE 14.2 (Stata Corporation, College Station, Texas). In addition, classification and regression tree (CART) analysis (performed in R [R Foundation for Statistical Computing, Vienna, Austria]) and linear discriminant analysis effect size (LEfSe) [34] were used to determine whether specific taxa differed significantly in relative abundance between women born by C-section or vaginal delivery. A heatmap, also created in R, included the 25 most abundant bacterial taxa found in the samples and were color-coded based on the relative abundance of each taxa in the samples.

## Ethics approval and consent to participate

The described research was performed in accordance with the Declaration of Helsinki and was approved by the institutional review boards of the University of Maryland College Park (reference #999286), University of Maryland Baltimore (HP-00073692 & HM-HP-00040935-14), and Johns Hopkins University (NA_00043112/CIR00024424). All participants provided either oral or written informed consent prior to completing phone or online surveys.

## Results

A total of 144 women were enrolled in this analysis, 88 from the HCL study and 56 from the VM400 study. At the time of sampling (parent study enrollment), the average age was 27.5

**Table 1. Characteristics of study participants (N = 144).**

| | C-Section (n = 27) | | Vaginal delivery (n = 117) | | |
|---|---|---|---|---|---|
| | n | % | n | % | P-value[e] |
| **Race/ethnicity** | | | | | 0.99 |
| Black or Latina | 9 | 33.3 | 40 | 34.2 | |
| Non-black and non-Latina | 18 | 66.7 | 77 | 65.8 | |
| **Age at study entry** | | | | | 0.66 |
| 17 to 23 | 10 | 37.0 | 33 | 28.2 | |
| 24 to 30 | 10 | 37.0 | 51 | 43.6 | |
| 31 & over | 7 | 25.9 | 33 | 28.2 | |
| **Body mass index [a]** | | | | | 0.03 |
| ≤24.9 | 16 | 59.3 | 58 | 50.0 | |
| 25.0–29.9 | 2 | 7.4 | 31 | 26.7 | |
| ≥30.0 | 9 | 33.3 | 27 | 23.3 | |
| **Community state type** | | | | | 0.51 |
| CST-I, *L. crispatus*-dominated | 11 | 40.7 | 45 | 38.5 | |
| CST-II, *L. gasseri*-dominated | 1 | 3.7 | 2 | 1.7 | |
| CST-III, *L. iners*-dominated | 8 | 29.6 | 38 | 32.5 | |
| CST-IV, Low *Lactobacillus* | 7 | 25.9 | 23 | 19.7 | |
| CST-V, *L. jensenii*-dominated | 0 | 0.0 | 9 | 7.7 | |
| **Bacterial vaginosis diagnosis within 2 months** | | | | | 0.61 |
| Yes | 1 | 3.7 | 6 | 5.1 | |
| No | 26 | 96.3 | 111 | 94.9 | |
| **Vaginal pH [b]** | | | | | 0.34 |
| 4.0–4.5 | 18 | 66.7 | 83 | 72.8 | |
| 4.6–5.0 | 3 | 11.1 | 18 | 15.8 | |
| >5.0 | 6 | 22.2 | 13 | 11.4 | |
| **Vaginal symptoms in prior 2 months** | | | | | |
| Discharge | 4 | 14.8 | 15 | 12.8 | 0.76 |
| Itching | 2 | 7.4 | 4 | 3.4 | 0.31 |
| **Ever been pregnant** | | | | | 0.53 |
| No | 17 | 63.0 | 66 | 56.4 | |
| Yes | 10 | 37.0 | 51 | 43.6 | |
| **Parity** | | | | | 0.33 |
| 0 | 18 | 66.7 | 79 | 67.5 | |
| 1 | 2 | 7.4 | 19 | 16.2 | |
| 2+ | 7 | 25.9 | 19 | 16.2 | |
| **Ever given birth vaginally** | | | | | 0.58 |
| No | 20 | 74.1 | 87 | 74.4 | |
| Yes | 7 | 25.9 | 30 | 25.6 | |
| **Years since last pregnancy [b]** | | | | | 0.75 |
| Never pregnant | 17 | 65.4 | 66 | 57.4 | |
| 3 or less | 4 | 15.4 | 23 | 20.0 | |
| more than 3 | 5 | 19.2 | 26 | 22.6 | |
| **Age at menarche [d]** | | | | | 0.18 |
| ≤12 | 18 | 66.7 | 59 | 50.4 | |
| >12 | 8 | 29.6 | 54 | 46.2 | |
| **Weight status at menarche [c]** | | | | | 0.49 |

(*Continued*)

**Table 1.** (Continued)

| | C-Section (n = 27) | | Vaginal delivery (n = 117) | | |
|---|---|---|---|---|---|
| | n | % | n | % | P-value[e] |
| Average or below | 23 | 85.2 | 93 | 79.5 | |
| Overweight or above | 4 | 14.8 | 20 | 17.1 | |
| **Hormonal contraceptive use (current)** | | | | | 0.54 |
| No | 14 | 51.9 | 62 | 53.0 | |
| Yes | 13 | 48.1 | 55 | 47.0 | |
| **Number of sexual partners in the prior 2 months [c]** | | | | | 0.55 |
| None | 7 | 25.9 | 19 | 16.8 | |
| 1 | 19 | 70.4 | 89 | 78.8 | |
| 2+ | 1 | 3.7 | 5 | 4.4 | |
| **Douched (ever)** | | | | | 0.03 |
| No | 17 | 63.0 | 98 | 83.8 | |
| Yes | 10 | 31.0 | 19 | 16.2 | |
| **Hygiene product use (2 months)** | | | | | |
| Feminine towellette | 1 | 3.7 | 11 | 9.4 | 0.47 |
| Hygiene spray | 0 | 0.0 | 3 | 7.0 | 0.99 |
| Hygiene powder | 1 | 8.3 | 2 | 4.8 | 0.54 |
| Other product | 2 | 7.4 | 10 | 8.6 | 0.60 |
| **Sanitary product use at last menstrual period [a]** | | | | | 0.15 |
| Tampon only | 8 | 30.8 | 37 | 31.6 | |
| Sanitary napkin only | 3 | 11.5 | 33 | 28.2 | |
| Tampon and sanitary napkin | 15 | 57.7 | 47 | 40.2 | |

a. Missing for 1 participant

b. Missing for 3 participants

c. Missing for 4 participants

d. Missing for 5 participants

e. P-values obtained from chi-square or Fisher's exact tests

years (standard deviation [SD] = 5.4, range = 17 to 44), 57.6% (n = 83) were nulliparous, and 9.2% (n = 13) reported no past sexual partners. No women reported current menstruation at the time of sampling, although one HCL woman reported that her enrollment visit coincided with the last day of her menstrual period. The median number of days since the end of the last menstrual period prior to sampling was 15 days in both studies.

Twenty-seven (18.8%) women reported birth by C-section. Women born by C-section were more likely to report higher BMI ($\geq$ 30 kg/m$^2$) and lifetime douching (both p < 0.05, Table 1). Women born by C-section and vaginal delivery were similar in terms of other demographic, behavioral, or health characteristics, including days since last menstrual period at time of sampling. However, there were significant differences between the former HCL and VM400 participants (S1 Table). Former HCL women were more likely to report younger age ($\leq$ 23 years) at enrollment, BMI of < 25 kg/m$^2$, past douching, using tampons only at last menstrual period, and younger age ($\leq$ 12 years) at menarche (all p<0.05). Though not significantly associated when categorized (p = 0.22), there was a moderate difference in the number of years since participants were last pregnant between HCL (median = 2.8 years) and VM400 (median = 4.7 years) women when assessed continuously (p = 0.10). No VM400 women reported vaginal discharge at enrollment (an exclusion criterion); however, they were more

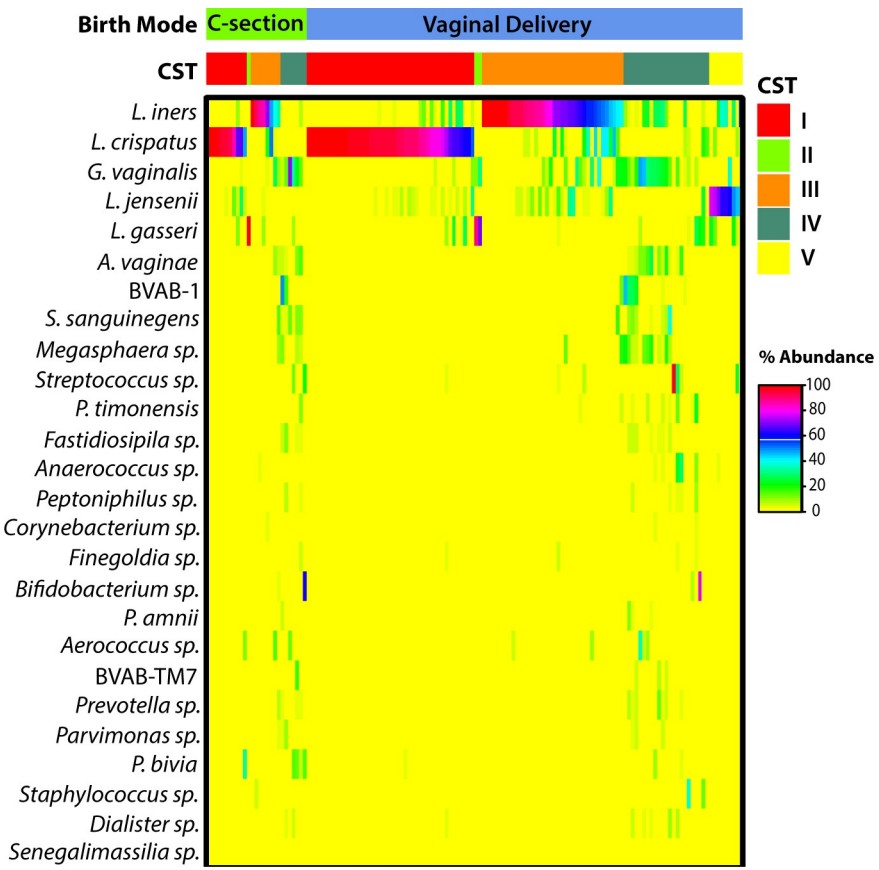

**Fig 1. Heatmap of bacterial relative abundance from cross-sectional sample previously collected from 144 women enrolled in the HCL and VM400 studies.**

likely to report vaginal itching or discharge in the prior two months compared to HCL women (both p < 0.05). The distributions of menstrual cycle phases were not statistically different between the studies (p = 0.40). In addition, ten (11.4%) HCL participants reported antibiotic use within the past two months of the baseline parent study visit (not statistically different between C-section- or vaginally-delivered groups), whereas antibiotic use within one month was an exclusion criterion for the VM400 study.

As previously reported [23], five CSTs were identified in this study population (Fig 1). Samples identified as Molecular-BV (CST IV), representing 20.8% (n = 30) of the study population, were compared to samples with high-*Lactobacillus* states (i.e., CST I, II, III, and V). Overall, birth mode was not significantly associated with Molecular-BV (CST IV); however, the OR estimate suggested increased odds of CST IV with C-section delivery (aOR = 1.22, p = 0.70, 95% CI: 0.45–3.32) (Table 2). BMI, the only variable that was associated with both C-section delivery and Molecular-BV, was the only confounder included in the adjusted models. Due to significant effect modification based on parent study site (interaction p = 0.048), we also stratified the analysis. In the HCL study stratum, women born by C-section had 3-fold higher odds of having low-*Lactobacillus* vaginal communities compared to vaginally-delivered women (n = 88, aOR = 3.55, p = 0.056, 95% CI: 0.97–13.02). No association was found within the VM400 study stratum (n = 56, aOR = 0.19, p = 0.14, 95% CI: 0.02–1.71). Sensitivity analyses excluding *L. iners*-dominated CST III from the high-*Lactobacillus* reference showed similar trends for the relationship between Molecular-BV and C-section. In a multinomial logistic

**Table 2. Association between C-section delivery and molecular-BV, combined (N = 144) and stratified by parent study.**

| Study | Birth Mode | Molecular-BV (n = 30) n (%) | *Lactobacillus*-dominated (n = 114) n (%) | Unadjusted | | Adjusted | |
|---|---|---|---|---|---|---|---|
| | | | | OR (95% CI) | P-value | aOR (95% CI) | P-value |
| HCL & VM400 | C-section (n = 27) | 7 (25.9) | 20 (74.1) | 1.43 (0.54, 3.79) | 0.47 | 1.22 (0.45, 3.32) | 0.70 |
| | Vaginal (n = 117) | 23 (19.7) | 94 (80.3) | 1 (ref) | | 1 (ref) | |
| HCL | C-section (n = 16) | 6 (37.5) | 10 (62.5) | 3.33 (1.00, 11.03) | 0.049 | 3.55 (0.97, 13.01) | 0.056 |
| | Vaginal (n = 72) | 11 (15.3) | 61 (84.7) | 1 (ref) | | 1 (ref) | |
| VM400 | C-section (n = 11) | 1 (9.1) | 10 (90.9) | 0.28 (0.03, 2.38) | 0.24 | 0.19 (0.02, 1.71) | 0.14 |
| | Vaginal (n = 45) | 12 (26.7) | 33 (73.3) | 1 (ref) | | 1 (ref) | |

Crude and adjusted odds ratios (ORs and aORs), 95% confidence intervals (CIs), and p-values were calculated using logistic regression. Percentages shown are row percents. Adjusted model controlled for BMI (categorized ≤24.9 [ref], 25.0–29.9, & ≥30). Molecular-BV (CST IV) was compared to *Lactobacillus*-dominated states (CST I, II, III, & V).

regression analysis of HCL study participants, the odds of having vaginal microbiota characterized as CST III was not associated with C-section (aOR = 1.29, p = 0.51, 95% CI: 0.31–5.22) and the odds of having CST IV retained its magnitude of association with C-section (aOR = 3.91, p = 0.06, 95% CI: 0.95–16.23) compared to other *Lactobacillus*-dominated CSTs.

Phylotype-level analyses confirmed that C-section delivery was not associated with high abundance of *Lactobacillus* species. In CART modelling, *Prevotella bivia* was the taxon with the strongest association with birth mode, and C-section-delivered women were more likely to have greater relative abundance (≥ 2.8%) of this bacteria (Fig 2A). High relative abundance of *P. bivia* was also associated with C-section in the LEfSe analysis; however, this result was non-significant (p = 0.23; Fig 2B). LEfSe also suggested that greater abundances of *L. jensenii* and *L. iners* were non-significantly associated with vaginal delivery.

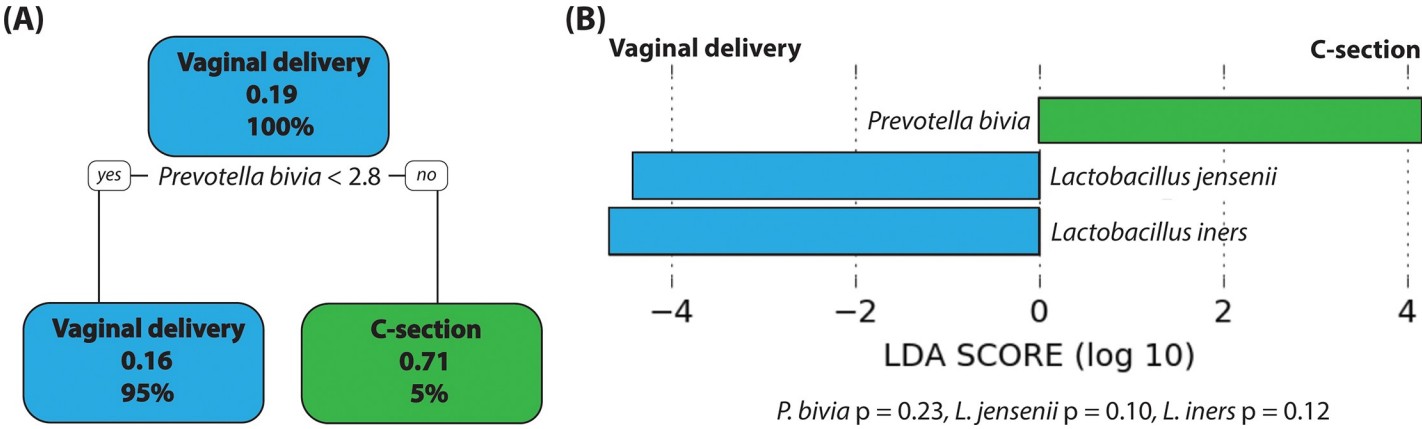

**Fig 2. Phylotype-level analyses results.** (A) In the pruned classification tree, each node shows the most likely birth mode (C-section or vaginal) given the bacterial relative abundance, the probability of C-section delivery in that node, and the percentage of observations in the node. *P. bivia* relative abundance was the taxon with the strongest association with birth mode history. If the relative abundance of *P. bivia* was less than 2.8%, the probability of birth mode being C-section was 16%. If the relative abundance of *P. bivia* was greater than 2.8%, the probability of birth mode being C-section-delivered was 71%. (B) The bacterial taxa with the highest effect sizes (linear discriminant analysis [LDA] scores > 4) reflect marked abundance in one birth mode group and not in the other. Three taxa were differentially abundant at this level, with *P. bivia* being more abundant in C-section-delivered group and *L. jensenii* and *L. iners* being more abundant in the vaginally-delivered group. However, these differences were not statistically significant (0.05 < p < 0.30).

## Discussion

In this cross-sectional pilot study of 144 reproductive-age women, we found modest evidence to suggest that a woman's vaginal microbiota in adulthood is associated with her mode of birth history. Among women recruited from the HCL study, being born by C-section was associated with a 3-fold increase in the odds of having the low-*Lactobacillus* CST IV (Molecular-BV) at a single time point in adulthood. *P. bivia*, a species often found in CST IV, was the bacterial taxon most strongly associated with birth mode, with higher relative abundances more often observed among C-section-delivered women.

We found qualitative interaction between study site and birth mode in the pooled analysis, which led us to stratify results by parent study. As with all subgroup analyses, the stratified results should be interpreted with caution as there is high likelihood of spurious findings. In addition, the sample sizes within strata were small, leading to concern for inadequate power. However, the low p-value of the test for interaction indicated low likelihood that the subgroup differences were due to chance.[35] The subgroup differences that were observed could be linked to marked differences between the study sites in age at sampling, weight, personal hygiene behaviors, and recent antibiotic use. As HCL participants were younger in age and were less likely to have given birth vaginally, we could reasonably hypothesize that the effect of birth mode could be stronger in this population. While the swabs used between the two studies were the same, the sample processing and sequencing methods differed. Our group's recent paper comparing vaginal samples processed using the dual-indexing one-step and two-step library preparation methods on the Illumina MiSeq (one-step and two-step) and HiSeq (two-step only) platforms found complete within-subject agreement among the CST assignments from all three methods.[25] Therefore, we have evidence to suggest limited variability in the CST outcome measurements between studies. CST in this study was determined from one vaginal sample collected at baseline. Our group and others have confirmed that the vaginal microbiota is dynamic for many women and can change at various points in the menstrual cycle and in pregnancy.[33,36–40] Basing the overall CST assignment on one sampling day may have resulted in misclassification of the typical bacterial composition of the participants. However, longitudinal studies have also demonstrated that women tend to stay in the same plane of fluctuation over many months and possibly years of observation.[33,38] Women who are found in CST IV at a single time point tend to most often fluctuate between IV and the *L. iners*-dominated CST III, which may represent an enduring pattern that favors Molecular-BV.[33] In addition, from an epidemiologic perspective, if there was non-differential misclassification of CSTs, that would serve to dampen the point estimates toward the null.[41] A finding of higher likelihood of Molecular-BV with CST IV, even within a cross-sectional analysis, still provides evidence for concern and could be indicative of long-term risk for BV and STIs.[42] The characterization of the vaginal microbiota was based on relative abundances of bacterial taxa. Although absolute abundances estimating the total bacterial loads [43] would complement the relative abundance results, they were not calculated in this analysis. Absolute abundance data provided by species-specific qPCR would also allow for observation of rare taxa and their relationship to birth mode. Our phylotype-level results indicating a higher relative abundance of *P. bivia* among women born by C-section reiterates the Molecular-BV findings in the CST-level analysis. *Prevotella* spp. have been associated with both BV and increased inflammation. [44–48] Si et al. also reported that *Prevotella* is a heritable bacteria that is associated with genetic variants of pro-inflammatory cytokines (interleukin-5) and obesity risk.[49]

BMI was the only variable associated with both birth mode and CST in the pooled analyses, and thus it was the only confounder included in the adjusted model. There is evidence to suggest that overweight mothers are more likely to have overweight daughters.[50,51] This

mother-daughter correlation of BMI led us to interpret this variable as a proxy for the mother's BMI, and a mother's BMI is often related to her delivery mode.[52,53] In addition, Mueller et al. demonstrated that overweight or obese offspring were more likely to be born to obese mothers by C-section, compared to children born vaginally to normal weight mothers.[54]

Our study is limited by missing variables on important maternal characteristics that may act as confounders, such as maternal age, the composition of the mother's vaginal microbiota preceding delivery, and intrapartum antibiotic exposure among mothers who gave birth by vaginal delivery. We did not collect the indication for C-section delivery or whether the surgery was elective or emergent. While these, and likely other variables, are undoubtedly important potential confounders, it may be difficult for adult daughters to recall these details accurately. In addition, there were many missing values in the breastfeeding variable, prohibiting use of this variable in assessing confounding. Like birth mode, there are conflicting findings in the literature on the effect of breastfeeding on the gut microbiota,[19] so it is not certain whether this factor would influence our analysis. Though the inability to adjust for breastfeeding history and other birthing/infancy characteristics may indicate unresolved confounding, the finding that these factors are not reliably obtained from adult women through direct questioning will inform future studies.

Another limitation of this study was the lack of accounting for the sociodemographic characteristics of both the mother and adult daughter within the analysis. It is known that a mother's socioeconomic status (SES) can influence whether she gives birth by C-section or vaginal delivery.[55] Indicators of SES, such as education and income, are also linked to BV prevalence.[56] While the racial/ethnic breakdown for re-contacted participants was similar between studies, it is likely that there were differences in SES between former HCL and VM400 participants. Unfortunately, SES information was not collected from VM400 participants at enrollment in the parent study or during the follow-up. Future studies should consider incorporating SES factors when assessing the intergenerational determinants for increased BV risk.

Our study has several strengths, including the efficient combination of archived samples and primary data collection to answer a novel research topic. To our knowledge, there are no available data on vaginal microbiota in adulthood as it relates to birth mode. Overall, the efforts to re-contact past participants were successful. We re-engaged over 72% of the former HCL participants who consented to be contacted for future research. However, former VM400 participants were less responsive, with less than 15% participating in the follow-up. The time interval between the end of the parent study and re-contact for this study on birth mode was much greater for VM400 than for HCL. Many participant phone numbers and email addresses had been disconnected. Self-selection bias was a minor concern, as women who participated in multiple follow-ups for the parent studies have proven to be very motivated and may be different from women who refused to participate. Only 37.5% of VM400 participants were able to report whether or not they were breastfed, which hindered further analysis of this variable. However, participants showed greater confidence when answering all other items, including their birth mode, in phone and online questionnaires, and in the few instances when participants were unsure of an answer, they were prompted to confirm their answers with a family member. The adult women were uniformly very confident in reporting their birth mode; any misclassification is likely to be independent of the Molecular-BV outcome.

There is little data published on how the vaginal microbiome is initially seeded. The establishment of stable adult vaginal microbiota may rely on several early environmental factors. There is conflicting evidence on the extent to which the acquisition of infant microbiota at several body sites is determined by birth mode.[57,58] While babies born by C-section delivery do not transit through the environment of the mothers' vaginal bacteria, they still acquire human microbes shortly after birth, possibly through contact with the operating room

environment [59] as well as through breastfeeding, diaper changes, and other close contact. After infancy, the vaginal microbiome undergoes significant transitions. In early childhood, girls tend to be colonized by stable aerobic, anaerobic, and enteric bacteria.[60] Important findings by Hickey *et al*. suggest that the vaginal microbiota of girls begins to resemble those of adults (typically dominated by *Lactobacillus* spp.) before menarche, while girls are still in the early and middle stages of puberty.[61] It is thought that the composition and function of the vaginal microbiota change in puberty due to increased estrogen production.[62] However, it remains unclear how girls transition and colonize adult-like vaginal microbiota, and further studies are needed to confirm the hypothesis that any effect of birth mode on the vaginal microbiota persists through infancy and puberty to adulthood.

Despite these controversies, there is evidence to support the hypothesis that birth mode influences health later in life. Babies born by C-section have an increased risk of childhood-onset type 1 diabetes and obesity at age 3 years, outcomes that may be related to the gut microbiota.[2,6] Most longitudinal studies investigating the effect of birth mode and neonatal exposures have followed infants for 3 years or less. For example, Chu et al. described a cohort study of 81 mother and infant dyads in which multiple body sites, including stool, oral gingiva, nares, skin and vagina, were sampled for 6 weeks.[20] The authors observed minor variations in the neonates' microbiota community structure associated with C-section delivery in most body sites immediately after birth. However, while the infants' microbiota matured substantially at each body site, the authors reported no discernable differences in bacterial community structure within gut, nares, and oral cavity between infants delivered vaginally or C-section at 6 weeks of age. In a recent pilot study, Dominguez-Bello et al. reported that C-section infants who were seeded with vaginal secretions from their mother had gut, oral, and skin bacterial communities enriched in vaginal bacteria after 30 days, similar to infants born vaginally.[63] These studies did not collect infant vaginal samples to determine how the vaginal microbiota undergoes reorganization in early life, which underscored the need for longitudinal studies assessing the seeding of bacteria in an infant girl's vaginal microenvironment.

BV is highly recurrent, associated with a number of adverse reproductive health outcomes, and is not effectively treated or prevented by current therapies.[16,64] Birth mode may represent a potentially modifiable risk factor. Broadly, interventions to prevent BV may include reducing C-section rates, reseeding the infant girl microbiome, or probiotic therapies administered to C-section-born women at various phases of the lifespan. Conclusive evidence from interventional studies would provide the causal link between birth mode and the composition and structure of the vaginal microbiota.

In summary, this novel pilot study found a moderately significant association between the low-*Lactobacillus* Molecular-BV state and being born by C-section, which indicates that C-section delivery may be related to having a less protective microbiota in adulthood. Most covariates we analyzed, including race, factors of puberty, and personal hygiene behaviors, did not alter the strength of the relationship between adulthood CST and birth mode. However, BMI was shown to be associated with C-section delivery and CST in this study sample. Although this cross-sectional study of a relatively small convenience sample may be limited by unmeasured confounding, and we are unable to establish causality or characterize the temporal dynamics of the vaginal microbiota, results suggest that birth mode is associated with vaginal seeding in the infant vaginal microbiome that may persist to adulthood.

## Supporting information

**S1 Table.**
(DOCX)

## Author Contributions

**Conceptualization:** Christina A. Stennett, Typhanye V. Dyer, Xin He, Jacques Ravel, Khalil G. Ghanem, Rebecca M. Brotman.

**Data curation:** Courtney K. Robinson.

**Formal analysis:** Christina A. Stennett, Xin He.

**Investigation:** Christina A. Stennett.

**Resources:** Courtney K. Robinson, Jacques Ravel, Khalil G. Ghanem, Rebecca M. Brotman.

**Writing – original draft:** Christina A. Stennett, Typhanye V. Dyer, Rebecca M. Brotman.

**Writing – review & editing:** Xin He, Courtney K. Robinson, Jacques Ravel, Khalil G. Ghanem, Rebecca M. Brotman.

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
