## [Decision Letter · Decision Letter 0]

11 Oct 2019

PONE-D-19-22430

A cross-sectional study of birth mode and vaginal microbiota in reproductive-age women

PLOS ONE

Dear Dr Brotman,

Thank you for submitting your manuscript to PLOS ONE. After careful consideration, we feel that it has merit but does not fully meet PLOS ONE’s publication criteria as it currently stands. Therefore, we invite you to submit a revised version of the manuscript that addresses the points raised during the review process.

The reviewer's comments  are addressed below. 

The comments and opinions of the reviewers were quite diverse and even opposed. However the subject of your study merits attention. Indeed, your report may initiate the discussion and encourage other researcher to conduct well designed, longitudinal, studies to elucidate the questions and hypothesis you raised in the submitted manuscript.

The manuscript will need a major revision, we agree with reviewer 1 that the reported study lacks power and that various confounders were not taken into account.   We also want to remind you of the journal’s criterion “experiments must have been conducted rigorously, with appropriate controls and replication, sample sizes must be large enough to produce robust results and methods and reagents must be described in sufficient detail for another researcher to reproduce the experiments described”.

We ask you to carefully address the comments of the first reviewer. Take into consideration to increase the sample size, to perform additional statistical analyses and to assess if the different procedural methods may have introduced a study to study variance.   

We concur with reviewer 2 that a supplementary table presenting the study populations side by side may be helpful for the readers.

We would appreciate receiving your revised manuscript by 1st of December 2019. To enhance the reproducibility of your results, we recommend that if applicable you deposit your laboratory protocols in protocols.io, where a protocol can be assigned its own identifier (DOI) such that it can be cited independently in the future. For instructions see: http://journals.plos.org/plosone/s/submission-guidelines#loc-laboratory-protocols

We look forward to receiving your revised manuscript.

Kind regards,

Tania Crucitti

Academic Editor

PLOS ONE

**Journal Requirements:**

3. Our internal editors have looked over your manuscript and determined that it is within the scope of our The Microbiome Across Biological Systems Call for Papers. This collection of papers is headed by a team of Guest Editors for PLOS ONE: Zaid Abdo, Colorado State University, USA; Sanjay Chotrimall, Lee Kong Chian School of Medicine, Nanyang Technological University, Singapore; Noelle Noyes, University of Minnesotta, USA;  Pankaj Trivedi, Colorado State University, USA; and Thomas Dawson, A*STAR, Singapore. The Collection will encompass a diverse range of research articles about microbiomes and human health, the natural and built environment, and new technologies used to study microbiomes. Additional information can be found on our announcement page: https://collections.plos.org/s/microbiome. If you would like your manuscript to be considered for this collection, please let us know in your cover letter and we will ensure that your paper is treated as if you were responding to this call. If you would prefer to remove your manuscript from collection consideration, please specify this in the cover letter.

**Additional Editor Comments (if provided):**

You stated that all data are made available, however, you should clarify on how this will be in addition to the reads that you will deposit in the SRA achive.

You cite ref 22 when referring the HCL study, this reference is not easily accesible, please consider another reference or a website where the report can be accessed.

Under limitations of the study we also would  like you to discuss the use of relative versus absolute abundance data and possible impact on your results.

Reviewers' comments:

Reviewer's Responses to Questions

**Comments to the Author**

1. Is the manuscript technically sound, and do the data support the conclusions?

Reviewer #1: Partly

Reviewer #2: Yes

Reviewer #3: Yes

2. Has the statistical analysis been performed appropriately and rigorously? 

Reviewer #1: No

Reviewer #2: Yes

Reviewer #3: Yes

3. Have the authors made all data underlying the findings in their manuscript fully available?

Reviewer #1: No

Reviewer #2: No

Reviewer #3: Yes

4. Is the manuscript presented in an intelligible fashion and written in standard English?

Reviewer #1: Yes

Reviewer #2: Yes

Reviewer #3: Yes

5. Review Comments to the Author

Reviewer #1: In this paper by Stennett and colleagues, attempts are made to associate birth mode with vaginal microbiota in adulthood. As currently presented, there are a number of issues with the article, particularly in the study design and subsequent interpretation of data, that prevent the paper being acceptable for publication. While the hypothesis of the study is interesting- that birth mode may influence likelihood of development of BV-like vaginal microbiota later in life- the study design is insufficient to enable the primary research question to be addressed in any meaningful way. The authors themselves acknowledge a lack of power which prevents associations (positive or negative) to be determined. However, there are other major concerns, some of which I have outlined below.

- The introduction fails to provide sufficient background and consideration of major known influences of vaginal microbiota composition. Chiefly, hormonally driven mechanisms of vaginal microbiota shaping throughout a woman’s life span, the lasting impact of pregnancy on the vaginal microbiota, menses etc. These are particularly important given the cross-sectional design and low power of the study, which makes the results highly prone to type I and II errors.

- While the multinomial logistic regression analyses provided a useful way adjust for 1 confounder (BMI), from what can be pertained from the methods, other likely confounders were not assessed and therefore could not be controlled for. For example, when in the menstrual cycle were base line samples for both cohorts collected? Were women asked if they had been recently pregnant? The latter is particularly pertinent given that women born by C-section were more likely to report giving birth within one year prior to enrolment in the parent study. Work from David Relman’s group and others have shown that pregnancy has a lasting impact on vaginal composition and is associated with a strong shift towards a “molecular BV” like composition for more than a year in some women. Thus, it is possible that the trend increase in molecular-BV type vaginal communities associated with delivery by C-section is due to a higher proportion of these women giving birth recently.

- Large amounts of missing data on breastfeeding at the time of sampling (72.5% missing) is also a major limitation. It is too superficial to discard its importance on vaginal microbiota composition by comparing it to the “conflicting findings in the literature on the effect of breastfeeding on the gut microbiota” as done so in the discussion. Breast feeding status would influence both hormonal levels as well as recommencement of menstruation and thus likely be an important confounder.

- Parts of the Methods are unclear and poorly described. For example, what is meant by “Taxa present at less than 10-5 across all samples were removed….”. Are you describing percentage of total read counts here?

- Given the difference in extractions, swabs used, PCR (one-step v two-step) sequencing platforms etc between the VM400 and HCL cohorts, how was study-to-study variance in resulting microbiota composition assessed? Where long-term reference samples used? If not, some samples should be split and analysed using both extraction and sequencing protocols side by side to ensure no bias was inadvertently introduced.

- Why were both chi-square and Fisher’s exact tests used? What were the assumptions for both?

- Was clustering and subsequent assignment of CSTs performed using the 25 most abundant bacterial taxa or all taxa?

- What are the “factors of puberty” that were considered in the adjusted models?

- Race and ethnicity seem to be used interchangeably throughout and it appears as though these terms were not used consistently across the VM400 and HCL cohorts.

- It is erroneous to suggest, as done so in the abstract and elsewhere in the paper, that Prevotella bivia or “molecular BV” are predictors of C-section. Care should be taken to not confuse interpretations of statistical testing with prediction of an event that occurred in the past (c-section)!

- The premise that birth mode may have a lasting impact on vaginal microbiota composition in later life demands more careful consideration and analysis of underlying clinical phenotypes at birth. While birth mode likely impacts early colonisation events of the vagina, other key exposures and events around the time of birth will also have a major influence (e.g. antibiotic exposure at time of delivery, being born prematurely, breastfeeding, etc). Without such detailed consideration, the findings presented herein remain spurious and unfortunately, not supported by the data presented.

Reviewer #2: Stennett et al have produced an interesting study that addresses a question that surprisingly has not been previously been looked at: whether differences in the vaginal microbiome stem from a woman's early life exposure to her mother's microbiota at the time of birth. The authors recontacted, and with full informed consent, surveyed women who had previously participated in two cross-sectional cohort studies (HCL and VM400) of the vaginal microbiome to determine the mode by which they were birthed. The study's primary analysis combined the cohorts to achieve greater statistical power, but each cohort was also examined separately. Multivariate logistic regression analysis found a non-statistically significant trend toward Molecular-BV with Cesarean section delivery, a trend that appeared to be driven largely by the HCL cohort. The last finding of interest in the analysis of individual bacterial taxa was that P. bivia was the best predictor of Cesarean birth.

This manuscript is well crafted and the analyses are appropriately conducted. The authors were cautious in their interpretation of the results and included a thorough discussion of the strengths and limitations of their study.

I recommend acceptance of the manuscript and offer a few minor recommendations to strengthen the manuscript, though I do not feel the manuscript's acceptance needs to be conditioned upon these modifications:

1) A reference that seemed to be missing and relevant is the Cell H&M paper by Si et al (2016) that described vaginal Prevotella as being a highly heritable. It is not clear at this point whether the P. bivia in the vaginal microbiotas of women born by Cesarean came from their mothers, but it is worth considering potential mechanisms that could support such a link.

2) As I was reading I frequently went looking for a table that described the two cohorts separately. It would have been nice to examine how relevant factors such as race and parity segregated across the cohorts in tabular format. Though a good amount of information was in the text, one's own curiosity and expediency of reference would be facilitated by a supplemental table.

3) Given a growing appreciation that having given birth has a long-lasting effect on the vaginal microbiome (MacIntyre et al. 2015 Scientific Report), I feel strongly that parity should be included in Table 1. It does not matter so much whether a woman has herself delivered vaginally or by Cesarean, but more whether she has had ever delivered a baby at term, or not.

4) Line 151 - was >> were

Reviewer #3: The current study represents an important step further in understanding the occurrence of BV. The field is slowly growing and the effect of C-section has been linked to number of serious health complications, but indeed as the authors point in their study not to much effort has been made to understand the effect of mode of delivery on the initial vaginal colonization and how this can affect women's life later. This makes the current study an interesting starting point to better understand the role of C-section and will clearly give the opportunities for longitudinal studies in the future.

The manuscript is clear, well organized and the the perform analysis is well selected.

I have only a few minor suggestions to the authors:

1. Discussion is rather long section, so I would advice the authors to shorter it a bit

2. Figures are not very clear, so authors should consider uploading figures with  higher resolution

6. PLOS authors have the option to publish the peer review history of their article (what does this mean?). If published, this will include your full peer review and any attached files.

Reviewer #1: No

Reviewer #2: No

Reviewer #3: No

---

## [Author Response · Author response to Decision Letter 0]

10 Jan 2020

We uploaded this response to reviewers with formatting as well. 

We thank the reviewers for their careful review and detailed feedback. The reviewers’ suggestions have led to significant improvements in this revision. Comments and response to reviewers are listed below. 

Reviewer #1: In this paper by Stennett and colleagues, attempts are made to associate birth mode with vaginal microbiota in adulthood. As currently presented, there are a number of issues with the article, particularly in the study design and subsequent interpretation of data, that prevent the paper being acceptable for publication. While the hypothesis of the study is interesting- that birth mode may influence likelihood of development of BV-like vaginal microbiota later in life- the study design is insufficient to enable the primary research question to be addressed in any meaningful way. The authors themselves acknowledge a lack of power which prevents associations (positive or negative) to be determined. However, there are other major concerns, some of which I have outlined below.

Response: A limitation of our research is the relatively low participation in the follow-up survey among former VM400 participants. However, we believe the study still provides important information when interpreted in context. While this convenience sample is not the definitive study on this topic, this novel work will serve as the impetus for further research and will inform future study designs and sample size calculations. In the revision, we refer to the analysis as a “pilot” study. 

Revisions: “A cross-sectional pilot study of birth mode and vaginal microbiota in reproductive-age women” Title

“Therefore, the purpose of this pilot study was to assess whether birth mode was associated with the composition of the vaginal microbiota of adult women in two study populations.” Introduction, line 74

“In this cross-sectional pilot study of 144 reproductive-age women, we found modest evidence to suggest that a woman’s vaginal microbiota in adulthood is associated with her mode of birth history.” Discussion, line 257

“In summary, this novel pilot study found a moderately significant association between the low-Lactobacillus Molecular-BV state and being born by C-section, which indicates that C-section delivery may be related to having a less protective microbiota in adulthood.” Discussion, line 389

1) The introduction fails to provide sufficient background and consideration of major known influences of vaginal microbiota composition. Chiefly, hormonally driven mechanisms of vaginal microbiota shaping throughout a woman’s life span, the lasting impact of pregnancy on the vaginal microbiota, menses etc. These are particularly important given the cross-sectional design and low power of the study, which makes the results highly prone to type I and II errors.

Response: In the introduction, we led with the public health significance of our study, describing research on long-term health effects of C-section delivery, and prior investigations on direct seeding of mothers’ microbiota to infants. In addition, some of the life course context is provided in the discussion—research characterizing the vaginal microbiota in early childhood (line 366) and puberty (line 361), as well as the longitudinal dynamics of the microbiota with respect to menstruation and pregnancy in reproductive-age (line 284). In the revised introduction section, we now introduce the concept that the effect of birth mode must persist through life course transitions. 

Revisions: “Thus, any differences in the composition of vaginal microbiota attributable to birth mode must persist through known hormonally-driven transitions in the microbiota during early childhood and puberty, including the longitudinal dynamics observed among reproductive-age women in menstruation and pregnancy.” Introduction, line 65 

2) While the multinomial logistic regression analyses provided a useful way adjust for 1 confounder (BMI), from what can be pertained from the methods, other likely confounders were not assessed and therefore could not be controlled for. For example, when in the menstrual cycle were base line samples for both cohorts collected? Were women asked if they had been recently pregnant? The latter is particularly pertinent given that women born by C-section were more likely to report giving birth within one year prior to enrolment in the parent study. Work from David Relman’s group and others have shown that pregnancy has a lasting impact on vaginal composition and is associated with a strong shift towards a “molecular BV” like composition for more than a year in some women. Thus, it is possible that the trend increase in Molecular-BV type vaginal communities associated with delivery by C-section is due to a higher proportion of these women giving birth recently.

Response: Menstruation at the time of sampling was an exclusion criterion for the VM400 study, and the former VM400 participants included in this analysis were sampled between 4 and 30 days (median=15, IQR=9 - 22) after their last menstrual period ended. In contrast, six out of 88 former HCL participants reported fewer than 4 days since last their last menstrual period (with 1 participant reporting that her period ended on the same day as her enrollment visit), and 6 participants reported greater than 40 days since last period at the enrollment visit (median=15.5, IQR=9 – 25). Several women in the HCL study were taking long-acting reversible contraceptives (LARCs), which affect regular menstrual cycles and contributed to the greater variability in this study population. 

We assessed menstrual cycle continuously and categorized at several different cut points and found no significant differences by birth mode, Molecular-BV status, or parent study. In the revision, we noted in the results section that our analyses did not appear to be affected by menstrual cycle phase. 

The years since last pregnancy variable was included in Table 1 in the previous submission. This variable was not associated with Molecular-BV in this sample; however, a greater proportion of women in the vaginal delivery group reported less than 3 years since last pregnancy compared to C-section-born women. In the revision, we discuss moderate differences in this variable between parent studies. 

Revisions: “No women reported current menstruation at the time of sampling, although one HCL woman reported that her enrollment visit coincided with the last day of her menstrual period. The median number of days since the end of the last menstrual period prior to sampling was 15 days in both studies.” Results, line 188

“Women born by C-section and vaginal delivery were similar in terms of other demographic, behavioral, or health characteristics, including days since last menstrual period at time of sampling.” Results, line 194

“Though not significantly associated when categorized (p = 0.22), there was a moderate difference in the number of years since participants were last pregnant between HCL (median = 2.8 years) and VM400 (median = 4.7 years) women when assessed continuously (p = 0.10). No VM400 women reported vaginal discharge at enrollment (an exclusion criterion); however, they were more likely to report vaginal itching or discharge in the prior two months compared to HCL women (both p < 0.05). The distributions of menstrual cycle phases were not statistically different between the studies (p = 0.40).” Results, line 199

3) Large amounts of missing data on breastfeeding at the time of sampling (72.5% missing) is also a major limitation. It is too superficial to discard its importance on vaginal microbiota composition by comparing it to the “conflicting findings in the literature on the effect of breastfeeding on the gut microbiota” as done so in the discussion. Breast feeding status would influence both hormonal levels as well as recommencement of menstruation and thus likely be an important confounder.

Response: Low participation in the breastfeeding survey question is a limitation. However, we believe the data remain useful to inform future study designs. Adult participants were less confident reporting their breastfeeding history compared to their birth mode history. Researchers who are considering study designs that ask adult women to self-report these factors can expect lower participation on some birthing/infancy survey questions than others. Though incomplete variables may lead to analytic limitations, prospective studies that follow participants from birth to adulthood are infeasible. 

Revision: “Though the inability to adjust for breastfeeding history and other birthing/infancy characteristics may indicate unresolved confounding, the finding that these factors are not reliably obtained from adult women through direct questioning will inform future studies Discussion, line 316

4) Parts of the Methods are unclear and poorly described. For example, what is meant by “Taxa present at less than 10-5 across all samples were removed….”. Are you describing percentage of total read counts here?

Response: We revised this sentence to be clearer about the rationale behind and the criteria for removing taxa and samples with few reads in the sequencing dataset. 

Revision: “Taxa for which the total read count across all samples was less than 10-5 were regarded as likely contaminants and removed. Individual samples with fewer than 1,000 total bacterial reads were also removed from the analysis.” Methods, line 127

5) Given the difference in extractions, swabs used, PCR (one-step v two-step) sequencing platforms etc. between the VM400 and HCL cohorts, how was study-to-study variance in resulting microbiota composition assessed? Where long-term reference samples used? If not, some samples should be split and analysed using both extraction and sequencing protocols side by side to ensure no bias was inadvertently introduced.

Response: Our group recently published a comparison of sample processing and sequencing methods and reported that CST assignments were consistent across the cited methods. We have added this information to the discussion of inter-study differences in the revised manuscript.

Revision: “While the swabs used between the two studies were the same, the sample processing and sequencing methods differed due to advancements made in sequencing technology over time. Our group’s recent paper comparing vaginal samples processed using the dual-indexing one-step and two-step library preparation methods on the Illumina MiSeq (one-step and two-step) and HiSeq (two-step only) platforms found complete within-subject agreement among the CST assignments from all three methods.[25] Therefore, we have evidence to suggest limited variability in the CST outcome measurements between the studies.” Discussion, line 273

6) Why were both chi-square and Fisher’s exact tests used? What were the assumptions for both?

Response: To examine the association between two categorical variables, the chi-square test applies an approximation assuming the sample is large. The Fisher's exact test runs an exact procedure that is more appropriate for small sample sizes, especially when some of the expected cell counts in a 2x2 table are less than five. For example, there was only one C-section-delivered woman who was diagnosed with BV within two months prior to sampling. The Fisher’s exact test was more appropriate to assess the association between birth mode and prior BV diagnosis due to the low cell count.

7) Was clustering and subsequent assignment of CSTs performed using the 25 most abundant bacterial taxa or all taxa?

Response: The CST assignments were based on all taxa (excluding contaminants) identified in 13,000+ archived samples. The heatmap only shows the 25 most abundant taxa identified in the 144 samples included in this analysis.

8) What are the “factors of puberty” that were considered in the adjusted models?

Response: The puberty-related variables we considered were age and weight status at the first menstrual period. This has been clarified in the revision.

Revision: “Several confounders identified in the literature as being associated with birth mode and/or Molecular-BV were considered for inclusion in the final adjusted model, including race/ethnicity, age, body mass index (BMI) at the time of sampling, and age and weight status at the first menstrual period.” Methods, line 169.

9) Race and ethnicity seem to be used interchangeably throughout and it appears as though these terms were not used consistently across the VM400 and HCL cohorts.

Response: Thank you for pointing out this oversight. The instance in the methods where race was used without ethnicity has been corrected. Both VM400 and HCL collected this variable similarly; ethnicity (i.e., Hispanic/Latina heritage) was not collected separately from race in either study, which is more common in newer demographic questionnaires. 

Revision: “Several confounders identified in the literature as being associated with birth mode and/or Molecular-BV were considered for inclusion in the final adjusted model, including race/ethnicity, age, body mass index (BMI) at the time of sampling, and age and weight status at the first menstrual period.” Methods, line 169.

10) It is erroneous to suggest, as done so in the abstract and elsewhere in the paper, that Prevotella bivia or “molecular BV” are predictors of C-section. Care should be taken to not confuse interpretations of statistical testing with prediction of an event that occurred in the past (csection)!

Response: We agree that this wording obscures the implied direction of association and have revised the abstract, results, and discussion. 

Revisions: “In CART modelling, Prevotella bivia was the taxon with the strongest association with birth mode, and C-section-delivered women were more likely to have greater relative abundance (≥ 2.8%) of this bacteria (Fig 2A). High relative abundance of P. bivia was also associated with C-section in the LEfSe analysis; however, this result was non-significant (p = 0.23; Fig 2B). LEfSe also suggested that greater abundances of L. jensenii and L. iners were non-significantly associated with vaginal delivery.” Results, line 236

“(A) In the pruned classification tree, each node shows the most likely birth mode (C-section or vaginal) given the bacterial relative abundance, the probability of C-section delivery in that node, and the percentage of observations in the node. P. bivia relative abundance was the taxon with the strongest association with birth mode history.” Fig 2 caption, line 243

“P. bivia, a species often found in CST IV, was the bacterial taxon most strongly associated with birth mode, with higher relative abundances more often observed among C-section-delivered women.” Discussion, line 261.

11) The premise that birth mode may have a lasting impact on vaginal microbiota composition in later life demands more careful consideration and analysis of underlying clinical phenotypes at birth. While birth mode likely impacts early colonisation events of the vagina, other key exposures and events around the time of birth will also have a major influence (e.g. antibiotic exposure at time of delivery, being born prematurely, breastfeeding, etc). Without such detailed consideration, the findings presented herein remain spurious and unfortunately, not supported by the data presented.

Response: Certainly, there are considerable limitations to this secondary data analysis. However, we still consider the results of this pilot study to be informative as they provide a useful first look at a novel research topic.

Revision: “Although this cross-sectional study of a relatively small convenience sample may be limited by unmeasured confounding, and we are unable to establish causality or characterize the temporal dynamics of the vaginal microbiota, results suggest that birth mode is associated with vaginal seeding in the infant vaginal microbiome that may persist to adulthood.” Discussion, line 394

Reviewer #2: Stennett et al have produced an interesting study that addresses a question that surprisingly has not been previously been looked at: whether differences in the vaginal microbiome stem from a woman's early life exposure to her mother's microbiota at the time of birth. The authors recontacted, and with full informed consent, surveyed women who had previously participated in two cross-sectional cohort studies (HCL and VM400) of the vaginal microbiome to determine the mode by which they were birthed. The study's primary analysis combined the cohorts to achieve greater statistical power, but each cohort was also examined separately. Multivariate logistic regression analysis found a non-statistically significant trend toward Molecular-BV with Cesarean section delivery, a trend that appeared to be driven largely by the HCL cohort. The last finding of interest in the analysis of individual bacterial taxa was that P. bivia was the best predictor of Cesarean birth.

This manuscript is well crafted and the analyses are appropriately conducted. The authors were cautious in their interpretation of the results and included a thorough discussion of the strengths and limitations of their study.

I recommend acceptance of the manuscript and offer a few minor recommendations to strengthen the manuscript, though I do not feel the manuscript's acceptance needs to be conditioned upon these modifications:

1) A reference that seemed to be missing and relevant is the Cell H&M paper by Si et al (2016) that described vaginal Prevotella as being a highly heritable. It is not clear at this point whether the P. bivia in the vaginal microbiotas of women born by Cesarean came from their mothers, but it is worth considering potential mechanisms that could support such a link.

Response: We thank the reviewer for this valuable feedback. The results of this relevant paper have been added to the paper. 

Revision: “Prevotella spp. have been associated with both BV and increased inflammation.[41–45] Si et al. also reported that Prevotella is a heritable bacteria that is associated with genetic variants of pro-inflammatory cytokines (interleukin 5) and obesity risk.[46]” Discussion, line 296

2) As I was reading, I frequently went looking for a table that described the two cohorts separately. It would have been nice to examine how relevant factors such as race and parity segregated across the cohorts in tabular format. Though a good amount of information was in the text, one's own curiosity and expediency of reference would be facilitated by a supplemental table.

Response: We agree that this table would be beneficial to the reader. S1 Table with the breakdown of the covariates by Parent study has been included with this revised submission. 

Revision: “However, there were a few significant differences between the former HCL and VM400 participants (S1 Table).” Results, line 196

3) Given a growing appreciation that having given birth has a long-lasting effect on the vaginal microbiome (MacIntyre et al. 2015 Scientific Report), I feel strongly that parity should be included in Table 1. It does not matter so much whether a woman has herself delivered vaginally or by Cesarean, but more whether she has had ever delivered a baby at term, or not.

Response: We have added parity to our revised tables.

Revisions: See Tables 1 and S1.

4) Line 151 - was >> were

Response: We have made the correction.

Revision: “There were many missing values for breastfeeding history questions with only 37.5% of VM400 participants providing responses.” Methods, line 155

Reviewer #3: The current study represents an important step further in understanding the occurrence of BV. The field is slowly growing and the effect of C-section has been linked to number of serious health complications, but indeed as the authors point in their study not too much effort has been made to understand the effect of mode of delivery on the initial vaginal colonization and how this can affect women's life later. This makes the current study an interesting starting point to better understand the role of C-section and will clearly give the opportunities for longitudinal studies in the future.

The manuscript is clear, well organized and the perform analysis is well selected.

I have only a few minor suggestions to the authors:

1. Discussion is rather long section, so I would advise the authors to shorter it a bit.

Response: We looked for areas in the discussion with unnecessary details. For instance, we removed some detail in the strengths paragraph (starting on line 334).

Deletion: “In phone interviews, three women expressed being unsure about their answers initially, particularly their ages at menarche, and were given the opportunity to contact family members to confirm. All three re-contacted research staff to confirm their responses, with one woman changing her age at menarche response. Limited missing data on the questionnaires administered online (including few “I don’t know” responses and skips for items other than breastfeeding history) indicated that women who participated online were similarly confident in their answers.”

Revision: “However, participants showed greater confidence when answering all other items, including their birth mode, in phone and online questionnaires, and in the few instances when participants were unsure of an answer, they were prompted to confirm their answers with a family member.” Discussion, line 343

2. Figures are not very clear, so authors should consider uploading figures with higher resolution.

Revision: The revised versions of Figures 1 & 2 now have higher resolution. 

Additional Editor Comments:

1. You stated that all data are made available, however, you should clarify on how this will be in addition to the reads that you will deposit in the SRA archive.

Response: All VM400 survey and sequence data used in this manuscript are submitted to dbGaP. We are currently authorized by the Johns Hopkins School of Medicine IRB to release three baseline HCL variables: race, age, and hormonal contraceptive status to dbGaP. However, additional survey data are available directly from the PI (Ghanem). We are in the process of receiving dbGaP ascension numbers for HCL. 

Revision: “The VM400 data can be found at the National Center for Biotechnology Information (NCBI) Database of Genotypes and Phenotypes (dbGaP) under accession number phs001909.v1.p1. For the HCL study, dbGaP accession numbers are in process. Additional survey data and accession numbers (when available) can be accessed directly from the Principal Investigator (Ghanem).” Declarations, line 408 

2. You cite ref 22 when referring the HCL study, this reference is not easily accesible, please consider another reference or a website where the report can be accessed.

Response: The reference has been updated so that it is easier to access. 

Revision: “22. Tuddenham S, Ghanem K, Gajer P, Robinson C, Ravel J, Brotman R. P591 The effect of hormonal contraception on the vaginal microbiota over 2 years. Sex Transm Infect. 2019;95: A263–A264. doi:10.1136/sextrans-2019-sti.662” 

3. Under limitations of the study we also would like you to discuss the use of relative versus absolute abundance data and possible impact on your results.

Response: We have revised the discussion to address this point. 

Revision: “The characterization of the vaginal microbiota was based on relative abundances of bacterial taxa. Although absolute abundances estimating the total bacterial loads [43] would complement the relative abundance results, they were not calculated in this analysis. Absolute abundance data provided by species-specific qPCR would also allow for observation of rare taxa and their relationship to birth mode.” Discussion, line 292.

---

## [Editor Report · Decision Letter 1]

21 Jan 2020

A cross-sectional pilot study of birth mode and vaginal microbiota in reproductive-age women

PONE-D-19-22430R1

Dear Dr. Brotman,

We are pleased to inform you that your manuscript has been judged scientifically suitable for publication and will be formally accepted for publication once it complies with all outstanding technical requirements.

With kind regards,

Tania Crucitti

Academic Editor

PLOS ONE
---

## [Editor Report · Acceptance letter]

16 Mar 2020

PONE-D-19-22430R1 

A cross-sectional pilot study of birth mode and vaginal microbiota in reproductive-age women 

Dear Dr. Brotman:

I am pleased to inform you that your manuscript has been deemed suitable for publication in PLOS ONE. Congratulations! Your manuscript is now with our production department. 

With kind regards,

on behalf of

Dr. Tania Crucitti 

Academic Editor

PLOS ONE